

# Patterns of gene evolution following duplications and speciations in vertebrates

Kyle T. David, Jamie R. Oaks and Kenneth M. Halanych

Department of Biological Sciences, Auburn University, Auburn, AL, USA

## ABSTRACT

**Background:** Eukaryotic genes typically form independent evolutionary lineages through either speciation or gene duplication events. Generally, gene copies resulting from speciation events (orthologs) are expected to maintain similarity over time with regard to sequence, structure and function. After a duplication event, however, resulting gene copies (paralogs) may experience a broader set of possible fates, including partial (subfunctionalization) or complete loss of function, as well as gain of new function (neofunctionalization). This assumption, known as the Ortholog Conjecture, is prevalent throughout molecular biology and notably plays an important role in many functional annotation methods. Unfortunately, studies that explicitly compare evolutionary processes between speciation and duplication events are rare and conflicting.

**Methods:** To provide an empirical assessment of ortholog/paralog evolution, we estimated ratios of nonsynonymous to synonymous substitutions ($\omega = dN/dS$) for 251,044 lineages in 6,244 gene trees across 77 vertebrate taxa.

**Results:** Overall, we found $\omega$ to be more similar between lineages descended from speciation events ($p < 0.001$) than lineages descended from duplication events, providing strong support for the Ortholog Conjecture. The asymmetry in $\omega$ following duplication events appears to be largely driven by an increase along one of the paralogous lineages, while the other remains similar to the parent. This trend is commonly associated with neofunctionalization, suggesting that gene duplication is a significant mechanism for generating novel gene functions.

## INTRODUCTION

Homologous relationships between eukaryotic genes are typically categorized as either orthologous or paralogous. Orthologs are gene copies that arise through speciation events and paralogs are gene copies that arise through duplication events (Fig. 1). The distinction between orthologs and paralogs has important implications for molecular biology largely due to the assumption that orthologs maintain similarity over time as genes are expected to serve comparable roles in descendant species (*Koonin, 2005*; *Dolinski & Botstein, 2007*; *Studer & Robinson-Rechavi, 2009*). By contrast, a much wider range of fates are considered for paralogs (*Otto & Yong, 2002*; *Innan & Kondrashov, 2010*; *Hahn, 2009*). In his seminal work, *Ohno (1970)* hypothesized several possible fates for duplicated genes. Paralogs may maintain their ancestral function (conservation (*Ohno, 1970*)),

Corresponding author
Kyle T. David, kzd0038@auburn.edu

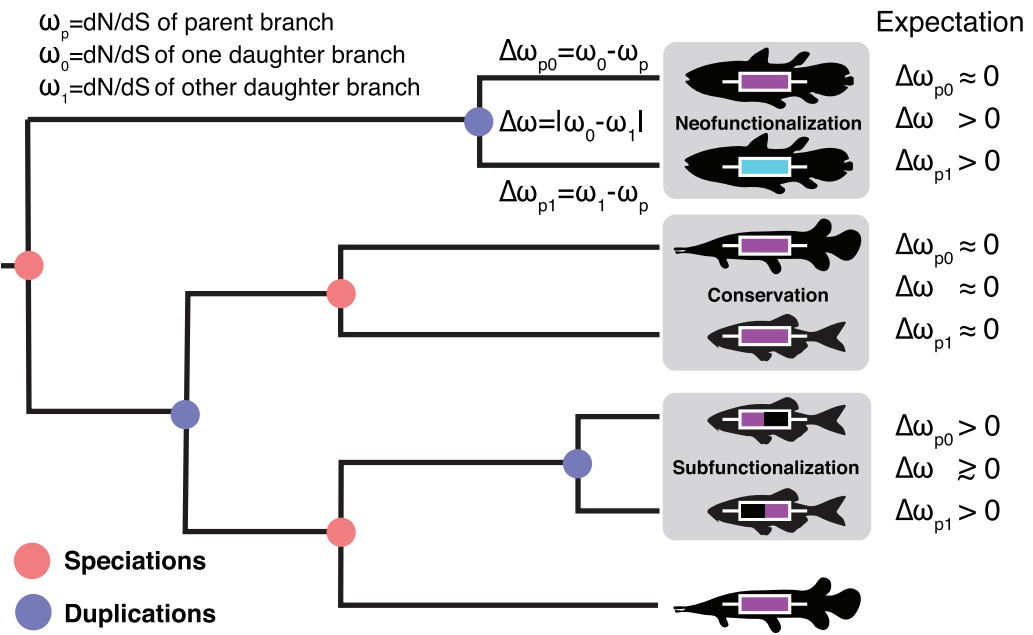

**Figure 1 The Ortholog Conjecture.** A hypothetical gene tree with duplication and speciation events. Genes that arose from a speciation event are orthologous to one another and genes that arose from a duplication event are paralogous. Under the Ortholog Conjecture, more changes are expected to occur along at least one of paralogous lineages following a duplication event. For each node we estimated the ratio of non-synonymous substitution rate (dN) over synonymous substitution rate (dS) for the two daughter lineages. We then calculated the absolute difference between the two ($\Delta\omega$) as well as the difference from the parent ($\Delta\omega_p$) for an estimate of divergence. Under neofunctionalization we would expect to see more nonsynonymous substitutions (positive $\Delta\omega_p$) in one lineage (whichever one is acquiring a new function) while the other lineage remains the same (small $\Delta\omega_p$), resulting in asymmetric selection between the two (large $\Delta\omega$). Under conservation, both lineages are expected to maintain similar levels of nonsynonymous substitutions as in the parent lineage (small $\Delta\omega_p$), resulting in symmetric selection (small $\Delta\omega$). Under subfunctionalization more nonsynonymous substitutions are predicted in both daughter lineages ($\Delta\omega_p$) as they partition aspects of the ancestral function. Subfunctionalization models generally do not make predictions regarding symmetry.

gain new function (neofunctionalization (*Force et al., 1999*)), divide or specialize the ancestral function between copies (subfunctionalization (*Force et al., 1999*)), or lose function entirely. Many additional models have since been proposed but which broadly fall under these main categories (*Innan & Kondrashov, 2010*; *Hahn, 2009*; *Lynch & Conery, 2000*; *Lynch & Katju, 2004*). These include escape from adaptive conflict (*Hughes, 1994*; *Des Marais & Rausher, 2008*), under which genes gain new function while continuing to perform the ancestral function as well as several dosage models, under which genes are preferentially retained or conserved on the basis of dosage effects.

The expectation of conservation between orthologs and divergence between paralogs is sometimes referred to as the "Ortholog Conjecture" (*Gabaldón & Koonin, 2013*; *Fitch, 1970*); an assumption so pervasive throughout genomics that it is not always referenced explicitly. For example, the Ortholog Conjecture is often used indirectly to infer gene function (*Dolinski & Botstein, 2007*; *Gabaldón & Koonin, 2013*). Under this method, the known function of a gene from *Mus musculus* would be assumed as the function for

orthologs in other species where the gene has not been functionally characterized. Many popular online databases such as KEGG (*Kanehisa et al., 2015*), PANTHER (*Thomas et al., 2003*) and eggNOG (*Huerta-Cepas et al., 2015*) rely on evidence from orthology to assign function. Additionally, as of October 2019, there are 386,841 proteins with orthology evidence in the Swiss-Prot sequence database representing 70% of all entries (*The UniProt Consortium, 2018*).

Given the importance of the Ortholog Conjecture in biology, the fact that only few studies test it explicitly (reviewed in *Studer & Robinson-Rechavi (2009)*, *Bleidorn (2017)*) is surprising. A preliminary study using microarrays between human and mouse recovered significant differences in expression profiles between ortholog pairs, at levels comparable with gene pairs selected at random (*Yanai, Graur & Ophir, 2004*). However, when these data were reanalyzed (*Liao & Zhang, 2005*), orthologs were found to be significantly more similar to one another than paralogs. *Nehrt et al. (2011)* used Gene Ontology annotations and found no support for the Ortholog Conjecture between humans and mice, however other researchers have since noted several pitfalls associated with using Gene Ontology (*Chen & Zhang, 2012*; *Altenhoff et al., 2012*; *Thomas et al., 2012*; *Rogozin et al., 2014*). Recently, a study by *Kryuchkova-Mostacci & Robinson-Rechavi (2016)* found strong support for the Ortholog Conjecture; however, reanalysis by *Dunn et al. (2018)* found the observed differences to be an artifact of node age, not the divergence events themselves.

To address the validity of the Ortholog Conjecture, we estimated the ratio of nonsynonymous to synonymous nucleotide substitution rates ($\omega = dN/dS$) for daughter lineages descended from inferred speciation events (orthologous lineages) or duplication events (paralogous lineages). We then took the absolute difference of $\omega$ ($\Delta\omega$) between daughter lineages for an estimate of the difference in selective pressure experienced by daughter lineages (Fig. 1). If nonsynonymous substitutions are more likely between paralogs, we would expect to see greater $\Delta\omega$ values following duplication events compared to speciation events (Fig. 1) (*Studer & Robinson-Rechavi, 2009*). We also measured the difference in $\omega$ of each daughter lineage from the parent ($\Delta\omega_p$) to see if one or both lineages diverged from the ancestral ratio. $\Delta\omega$ was estimated for 234,066 speciation events and 16,978 duplication events in 6,244 gene families across 77 vertebrates (Table S1).

## MATERIALS AND METHODS

We drew from 22,340 publicly available protein trees from the EnsemblCompara online database (release 90) (*Vilella et al., 2009*; *Herrero et al., 2016*). To avoid issues with unknown calibration dates and branch lengths, we focused on patterns within vertebrates for 77 target taxa and 6 outgroup taxa with a well-established evolutionary history (Table S1). Calibration dates taken from the most recent literature (Table S2) were assigned to 55 nodes in the species tree and implemented in a global clock model (*Yang, 2007*). Under the global clock model one rate is used for each tree, represented in millions of years. Calibrating the data in this way enables us to compare nodes from different trees to one another, even though they likely have different rates of substitution. Trees that lacked at least one node with a calibration date (9,528) were

discarded. Paralogs and orthologs were inferred using the species overlap algorithm (*Huerta-Cepas et al., 2007*). Duplication labels were applied to nodes where the same species is represented in both daughter clades at least once. Speciation labels were then those nodes with no species overlap between the two daughter lineages. Put differently, speciation nodes gave rise to two discrete monophyletic groups and duplication nodes did not (Fig. 1). Speciation and duplication labels generated from this method were compared with the annotations provided by EnsemblCompara (*Vilella et al., 2009*), with which they were congruent in 97.5% of cases. This study ignores horizontal gene transfer events and their resultant xenologs, which are unlikely between vertebrates.

Trees with greater than approximately 170 tips (~5,000 trees) were found to be too computationally intensive and excluded from our analyses. We also filtered 2,766 trees that did not contain at least one duplication and at least one speciation event. Of the remaining 6,303 trees, we filtered 31,039 nodes with an expected number of synonymous substitutions greater than 2 in either daughter lineage, which indicate saturation of substitutions, then 75,427 nodes with an expected number of synonymous substitutions less than 0.01 in either daughter lineage, which can lead to poor estimates of $\omega$, and finally 61 nodes with $\omega > 10$ in one or both daughter lineages as outliers (*Villanueva-Cañas, Laurie & Albà, 2013*). Our filtered dataset contains 6,244 trees with 16,978 duplication nodes and 234,066 speciation nodes.

We inferred selective pressure by estimating the rate of nonsynonymous substitutions relative to the rate of synonymous substitutions ($\omega$) (*Yang & Bielawski, 2000*). We estimated $\omega$ values with codeml, a maximum-likelihood method for codon-substitutions within the PAML package (*Yang, 2007*). To estimate $\Delta\omega$, we used a free-ratios model which allows separate $\omega$ values to be calculated for each branch in the tree (*Yang, 1998a*, *1998b*). The difference in selective pressure was then quantified by simply taking the absolute difference between the $\omega$ of the two daughter lineages ($\Delta\omega$) as well as the difference of one daughter lineage from the parent lineage ($\Delta\omega_P$) for each speciation and duplication node in our tree (Fig. 1). Per the PAML authors' recommendation we first ran a null model, which assumes uniform $\omega$ values to generate branch lengths and transition/transversion ratios in an effort to limit free parameters. All analyses were performed on the High Performance Computing Hopper Cluster at Auburn University.

To effectively test the Ortholog Conjecture we compared empirical values to a null model, in which there was no difference between speciation and gene duplication events. For our null model, we assigned speciation and duplication labels randomly without replacement for each gene tree, removing any putative link between evolutionary events and $\Delta\omega$. To calculate *p*-values, we performed a two-tailed permutation test comparing our empirical estimates to those calculated from trees with permuted speciation/duplication node labels. Null distributions were approximated with 1,000 permutations. Under either model, the number of speciation and gene duplication nodes for each tree was kept the same as in the empirical trees. We also include Hedges' *g* (*Hedges, 1981*) as a measure of effect size.

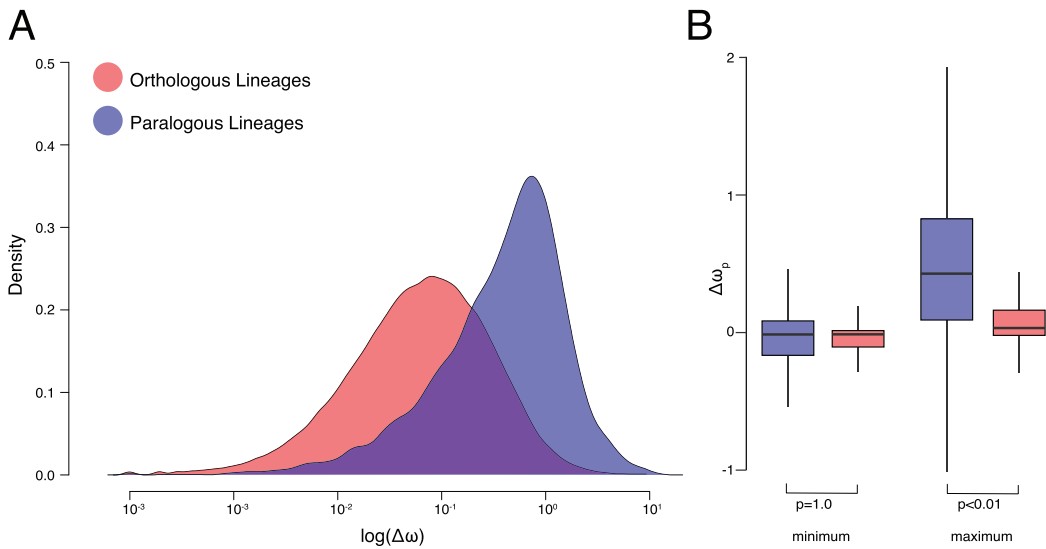

**Figure 2 ω Estimates for orthologs and paralogs.** (A) Kernel density plots of log transformed Δω of orthologous and paralogous lineages. (B) Δω$_P$ of orthologous and paralogous lineages with each pair of daughter lineages categorized by maximum and minimum values.

For our analysis, we were interested in testing not just differences between means but also similarity of distributions. Considering this, we calculated the overlap coefficient (OVL) between kernel density distributions of orthologous and paralogous lineages (Fig. 2A). The OVL is the area in common under two probability density functions and represents the sum of the conditional misclassification probability as well as providing an intuitive measure of agreement between two similar distributions (*Inman & Bradley, 1989*). Under the most extreme interpretation of the Ortholog Conjecture we would expect no overlap in Δω distributions between orthologous and paralogous lineages. OVLs were calculated by integrating the area under the intersection between the two density plots. We also perform a two-sample Kolmogorov–Smirnov test, which is a nonparametric test which estimates the likelihood of two samples (in this case Δω values between speciation and duplication events) being drawn from the same distribution (*Massey, 1951*).

All code required to update experiments and reproduce results/figures are available at https://github.com/KyleTDavid/OrthologConjecture2019. Original data files are available at https://figshare.com/projects/OrthologConjecture2019/63935.

## RESULTS

On average, Δω was significantly smaller between orthologous lineages than paralogous lineages ($p < 0.001$, K–S test $p < 2.2E{-}16$, Hedges' $g = 0.94$) (Fig. 2A). After a speciation event, resulting orthologs experience more similar patterns of molecular evolution (average Δω was $0.20 \pm 0.38$ standard deviations) to one another than do paralogs after a duplication event ($0.60 \pm 0.77$). The OVL of kernel density distributions of Δω was 57.6% between orthologous and paralogous lineages. Small but significant ($p < 0.001$, Hedges' $g = 0.18$) differences in Δω were also recovered between different categories of duplication

events, with greater $\Delta\omega$ between duplication events leading to within-species paralogs (i.e., nodes in which all descendant leaves belong to the same species) ($0.62 \pm 0.77$) than between-species paralogs (i.e., duplication nodes in which descendant leaves belong to more than one species) ($0.48 \pm 0.77$) (Fig. S1).

We observe significant ($p < 0.001$, Hedges' $g = 0.88$) differences in the $\omega$ ratios themselves with an average ratio of $0.22 \pm 0.36$ for orthologous lineages compared to $0.56 \pm 0.70$ for paralogous lineages (Fig. S2). High $\omega$ following gene duplications events is a well-documented phenomenon (*Lynch & Conery, 2000*; *Seoighe, Johnston & Shields, 2003*; *Johnston et al., 2006*; *Han et al., 2009*). However whether or not relaxed selective pressures are experienced equally between copies remains controversial (*Kondrashov et al., 2002*; *Scannell & Wolfe, 2008*). To address this we also estimated the minimum and maximum difference between $\omega$ of each daughter lineage per pair and their parent lineage ($\Delta\omega_p$) (Fig. 1). $\omega$ for paralogous lineages increased by $0.21 \pm 0.82$ compared to their parent lineage, significantly ($p < 0.001$, Hedges' $g = 0.48$) higher than $\omega$ for orthologous lineages which decrease by $0.003 \pm 0.42$. Although differences from the parent lineage were not recovered between the minimum orthologous and paralogous daughter lineages ($p = 1.00$) they were recovered for the maximum lineages ($p < 0.01$) (Fig. 2B). Additionally, effect sizes between the maximum and minimum lineages were >100× greater between paralogous (Hedges' $g = 0.85$) and orthologous lineages (Hedges' $g = 6.7E{-}3$).

## DISCUSSION

Our results demonstrate support for the Ortholog Conjecture with regard to $\omega$. The relative rates of nonsynonymous substitutions differs more between paralogous lineages than between orthologous lineages. Our observation of higher $\Delta\omega$ in paralogous lineages appears to be largely the result of a higher relative rate of nonsynonymous substitutions in just one daughter paralog (Fig. 2B). This pattern is commonly thought to be indicative of Ohno's neofunctionalization (*Ohno, 1970*) or Francino's adaptive radiation (*Francino, 2005*) models of evolution (*Innan & Kondrashov, 2010*; *Lynch & Katju, 2004*), indicating that diversification may be the most common fate for at least one retained paralog copy. This interpretation is congruent with previous studies (*Han et al., 2009*; *Conant & Wagner, 2003*; *Kellis, Birren & Lander, 2004*), such as *Brunet et al. (2006)* who also observed conserved vs. elevated selection between paralogous lineages resulting from the teleost specific whole genome duplication event. However, a later study (*Studer et al., 2008*) using a more rigorous model found evidence for high levels of selection across lineages throughout vertebrates, regardless of homologous relationships. It worth noting that this interpretation only extends insofar as relative rates of substitution can be understood as a proxy for functional change. In reality the relationship between sequence, structure, and function is far from direct. For example, there is also a negative correlation between $\omega$ and level of expression (*Drummond et al., 2005*). If observed $\omega$ values were driven by expression our results could also support a specialization model of subfunctionalization, under which the new copy continues to perform the ancestral function in a reduced, specialized capacity (expressed only in particular tissues, under certain environmental conditions, etc.).

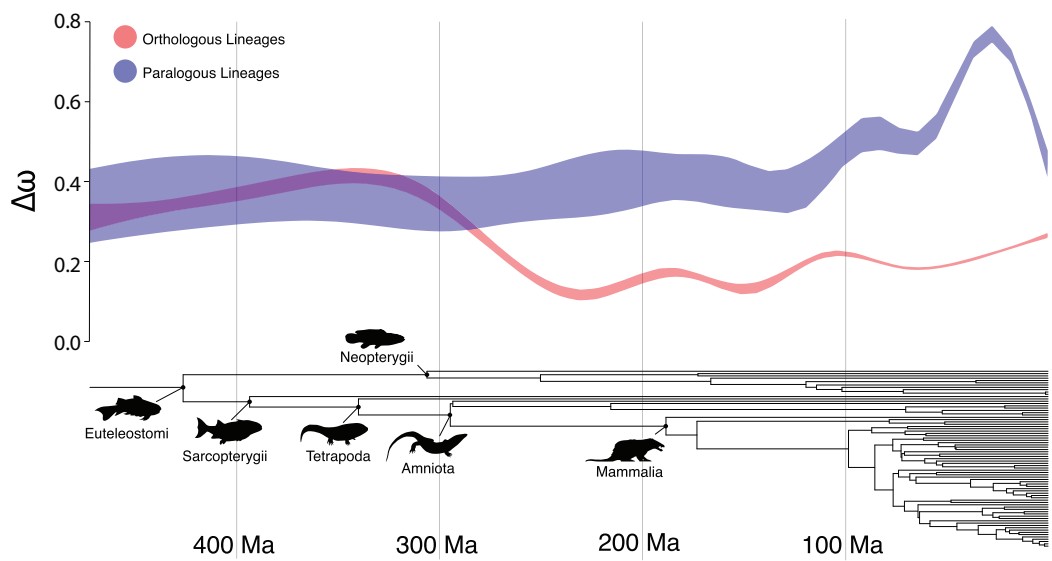

**Figure 3** **The Ortholog Conjecture through time.** General additive model 95% confidence intervals of Δω following speciation and duplication events over time, over the time-calibrated species tree of all taxa included in the study.

Most gene duplication events in our dataset occur along lineages for which there is no evidence for whole genome duplication (98.3%), indicating that the Ortholog Conjecture is as, if not more, pronounced in single/several gene duplication events (such as those produced through unequal crossing-over) than whole genome duplication events. New gene copies generated through whole genome duplication are identical in terms of just not their sequence but location and context within the (sub)genome as well (*Studer & Robinson-Rechavi, 2009*). By contrast retrotransposed gene duplicates are relocated to an entirely new genomic environment in which it will be unlikely to reproduce the ancestral function and thus free to acquire new mutations. This scenario is supported by *Han et al. (2009)* who found that gene duplicates transferred to a new position are more likely to experience elevated ω ratios.

There were several clades in which the Ortholog Conjecture was not supported ($p > 0.05$), namely: Sarcopterygii, Neopterygii, Amniota and Mammalia. This suggests an unexpected trend of decreasing support for the Ortholog Conjecture as time since divergence increases (Fig. 3). This trend is likely the result of saturation in substitutions (*Cannarozzi & Schneider, 2012*) over time as the expected number of synonymous substitutions begin to plateau at ~150 Ma while the expected number of nonsynonymous substitutions continue to increase, obscuring differences between duplication and speciation events and increasing the overlap between the distributions (Fig. S3). Additionally, estimates of ω have been demonstrated to exhibit slight bias when divergence levels are low (*Stoletzki & Eyre-Walker, 2010*). As a result, more work may need to be done in order to more definitively resolve questions of ortholog and paralog evolution along especially old or young vertebrate lineages.

A similar pattern was recovered between different classifications of paralogs, with slightly smaller Δω for between-species paralogs than within-species paralogs (Fig. S1).

This finding disagrees with previous results which suggest more conservation between within-species paralogs than between-species paralogs (*Nehrt et al., 2011*; *Altenhoff et al., 2012*). This discrepancy may be the result of saturation as noted above, as within-species duplication events are necessarily concentrated toward the tips of a phylogeny. However, when all nodes were filtered to those <1 Ma significant ($p < 0.01$) differences were still recovered.

Notably, our approach only estimates the expected average ω for each branch. This could be a source of error, particularly if functionally relevant mutations occur early or late in the history of a lineage (*Innan & Kondrashov, 2010*; *Han et al., 2009*). Similarly, our method is ignorant to models that involve multiple predictions at different stages of a paralog's history. For example, He and Zhang's subneofunctionalization model predicts paralogs undergo a short subfunctionalization period followed by sustained neofunctionalization (*He & Zhang, 2005*).

## CONCLUSION

Our results reveal distinct differences in patterns of substitution between lineages descending from speciation and duplication events. ω ratios are on average more similar between daughter lineages following speciation events than duplication events. This greater amount of asymmetric evolution between paralogs appears to be driven by an increase in relaxed selection in just one of the lineages. The maximum change from the parent branch is 5.6× greater for paralogous lineages than orthologous lineages, whereas the minimum change from the parent branch is nearly the same (0.98×) (Fig. 2B). Taken together these trends may indicate neofunctionalization, where one copy retains the ancestral function while the other acquires a new function, as the most dominant pattern for retained duplicated genes.

## ACKNOWLEDGEMENTS

We would like to thank members of the Molette and Phyletica labs, in particular Damien Waits, Caitlin Redak, Michael Tassia, and Dr. Yuanning Li for their feedback and comments. This is Molette Biology Laboratory contribution #99, Auburn University Marine Biology Program contribution #198, and Auburn University Museum of Natural History contribution #924.

### Funding

This project was supported by the AU Cell and Molecular Biosciences Fellowship, the NSF Graduate Research Fellowship (NSF-DGE-1937964), and the Schneller Endowed Chair Fund. The funders had no role in study design, data collection and analysis, decision to publish, or preparation of the manuscript.

## Grant Disclosures

The following grant information was disclosed by the authors:

AU Cell and Molecular Biosciences Fellowship.

NSF Graduate Research Fellowship: NSF-DGE-1937964.

Schneller Endowed Chair Fund.

## Competing Interests

The authors declare that they have no competing interests.

## Author Contributions

- Kyle T. David conceived and designed the experiments, performed the experiments, analyzed the data, prepared figures and/or tables, authored or reviewed drafts of the paper, and approved the final draft.
- Jamie R. Oaks conceived and designed the experiments, authored or reviewed drafts of the paper, and approved the final draft.
- Ken M. Halanych analyzed the data, authored or reviewed drafts of the paper, and approved the final draft.

## Data Availability

All code required to update experiments and reproduce results/figures are available at GitHub: https://github.com/KyleTDavid/OrthologConjecture2019. Original data files are available at Figshare: https://figshare.com/projects/OrthologConjecture2019/63935.

## Supplemental Information

Supplemental information for this article can be found online at http://dx.doi.org/10.7717/peerj.8813#supplemental-information.

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
