# Peer review of "Patterns of gene evolution following duplications and speciations in vertebrates"

_PeerJ, doi:10.7717/peerj.8813_

## Round 0.1 · original submission · Major Revisions

The expert referees have provided a number of recommendations, some of which overlap between multiple referees. The authors should make every effort to incorporate the the revisions required by all of these suggestions.

Reviewer 1 ·

Basic reporting

No comment

Experimental design

Review of David, Oaks, and Halanych: “Patterns of gene evolution following duplications and speciations in vertebrates.”

Overview:
In this manuscript, the authors describe having inferred a large set of gene trees that include branchings due both to speciation and duplication events. They divide the branches in this fashion and estimate branch-specific values of the selective constraint (Ka/Ks) for all branches. They then explore the pattern of shifts in Ka/Ks after speciation and duplication events. They argue that the pattern of increases in Ka/Ks after duplication events is consistent with the orthology conjecture: that orthologs maintain more similar functions over time than do gene duplicates.

Major comments:
The authors have done a commendable job of assembling a valuable and interesting set of genes for analysis. At the same time, I do not find that authors’ interpretation of their results at all convincing and think that both the methodology and that interpretation need additional analyses and background reading.

My primary concern is that I do not understand at all why the authors dismiss perhaps the most obvious explanation of their results. We expect a relaxation of selective constraint immediately after gene duplication, as is discussed in several of the papers cited here. In many cases, that relaxation will be followed by nonfunctionalization. However, even in surviving duplicates, the relaxation should leave at least some signature on the branch-wise Ka/Ks, even if the genes in question later come under a stronger selective regime (because you cannot “unfix” those neutral amino acid-changing substitutions from the early history of the duplicate.) To my mind, all of the authors’ results are consistent with this null model, without any need to involve the orthology conjecture, neofunctionalization or any other models of duplicate evolution. Now, I do not mean by this that no other forces are at work or that the orthology conjecture is wrong. I just do not find the data here as speaking to these questions.

I also think that the methods used would need some refinement to even support the conclusion of relaxed selection after duplication. The Ka/Ks ratio can have odd time-dependent behavior that make comparing old and young gene duplications difficult even aside from the complexity of understanding gene duplication (1). Likewise, although EggNog is a viable orthology inference method, I would like to see at least one more orthology approach (EnsemblCompara?) used in the analysis to be sure that the inference of orthology/paralogy nodes is not biasing any results.

Minor points:

It has been some time since I read Ohno’s book, but I do not recall him introducing the concept of subfunctionalization. Similarly, I think it is a mistake to claim that other mechanisms of duplicate preservation fall under the sub/neo umbrella: see especially (2-5).

Lines 112-113: I think it is clearer to describe this model as allowing different Ka/Ks ratios on each branch rather than each lineage.

Line 194-196: It is not correct to interpret either elevated Ka/Ks <1.0 or asymmetry in Ka/Ks as evidence of neofunctionalization: many of the works cited here make this point quite clear.


References:


1. Stoletzki N & Eyre-Walker A (2011) The positive correlation between dN/dS and dS in mammals is due to runs of adjacent substitutions. Molecular Biology and Evolution 28(4):1371-1380.
2. Kondrashov FA & Kondrashov AS (2006) Role of selection in fixation of gene duplications. Journal of Theoretical Biology 239(2):141-151.
3. Birchler JA & Veitia RA (2012) Gene balance hypothesis: connecting issues of dosage sensitivity across biological disciplines. Proc Natl Acad Sci U S A 109(37):14746-14753.
4. Freeling M (2009) Bias in plant gene content following different sorts of duplication: tandem, whole-genome, segmental, or by transposition. Annual Review of Plant Biology 60:433-453.
5. Makino T & McLysaght A (2010) Ohnologs in the human genome are dosage balanced and frequently associated with disease. Proceedings of the National Academy of Sciences, U.S.A. 107(20):9270-9274.

Validity of the findings

See #2

Additional comments

See above.

Reviewer 2 ·

Basic reporting

David et al investigate the differential functional divergence between ortholog and paralog gene lineages, as assumed by the Ortholog Conjecture (i.e. pairs of orthologous gene lineages are assumed to be less divergent than pairs of paralogous gene lineages, since they presumably preserve the same functionality).

They estimate differential functional divergence by first evaluating the ratio of non-synonymous to synonymous substitutions (dN/dS) in independent gene lineages deriving from either speciation events (orthologous lineages) or duplication events (paralogous lineages). Then, they evaluate the difference in dN/dS ratio between pairs of orthologous lineages and paralogous lineages, trying to test if/when there is a significant statistical difference between the two groups (orthologs vs paralogs).

Overall, we found the study to be clear and interesting, and a potentially useful resource as well. Therefore, we would recommend publication after minor changes.

Experimental design

no comment

Validity of the findings

no comment

Additional comments

Suggestions/comments:

* Introduction:

The use of the English language is quite clear, the text flows well and is easy to follow. Statements are in general well referenced, but they are perhaps missing the original reference for the “Ortholog Conjecture” (lines 51-53).

* Methods:

Line 92: Since calibration dates seem to be key, we suggest spending some more words on the global clock model with which they are determined (it is well referenced, though).

Line 109: Selection pressure == selective pressure?


* Results:

- Line 156: The authors test if there are significant differences between two categories of duplication events (paralogs): within-species and between-species. We wonder, are there differences between other categories of paralogs, e.g. 1:2, 1:3, etc? Or, perhaps more interesting, when there is a single duplication (1:2) vs. when there are more lineage amplifications (1:many).

- Line 158: “significant (p<0.001, Hedges’ g=0.18) differences in Δω were also recovered between different categories of duplication events”. Even if the difference is significant, it might be useful to stress that the effect size (as estimated by the Hedges’s g) is quite limited. It might also be good to have a figure to illustrate this result.

- Line 161: We wonder whether it would be possible to have a figure for this result as well, since this is one of the most important results.

- Lines 165 –175: These results seem to be consistent with those obtained for differently specialized families after the vertebrate whole genome duplications (Marletaz et al, Nature 2018).

- Will the raw values per gene and species be available as a Supplementary (text) Table? This would be a nice resource for the community.

* Discussion:

- Line 201: this paragraph should be more appropriately explained in the Results section (with relative description of Fig3 there as. well). Also, the sentence could be clearer, e.g.: “When Δω is estimated over time, two unexpected patterns emerge (Fig. 3). First, there is an apparent spike in Δω between paralogous lineages that diverged ~25 mya, with an average Δω of0.73 ± 0.83 for duplication events from between 10mya and 40mya”.


* Figures:

- Fig1. We have struggled a bit to understand this figure. Therefore, we recommend to modify it a bit to make it easier to follow. Perhaps relatedly, we found the caption incomplete: since the possible fates of paralogs are illustrated, it might be good to mention/comment them in the caption?

- Fig2. It might be better if it was divided in A and B and referenced as such.

- Fig3. The connection between upper and lower panel should be better explained in the caption.

·

Basic reporting

First, I have now understood why some values of Δω are always positive, and others are not. On Fig 1, it is indeed noted that Δω between daughter branches is calculated as an absolute value, whereas Δωp is not, and this is consistent with the R code. Except that in the R code, Δωp is used to call the function run(), which uses abs().
The authors need to (i) clarify whether Δωp is tested with absolute values, and (ii) make it clear in the text which Δω values can or cannot be negative, and why.

The authors continue to ignore previous work in their discussion, even though they are cited in passing. I repeat the references I provided last time, which performed very similar analyses and must be truly discussed:
Brunet et al 2006 https://academic.oup.com/mbe/article/23/9/1808/1014301
Studer et al 2008 https://genome.cshlp.org/content/18/9/1393.abstract
Han et al 2009 https://genome.cshlp.org/content/19/5/859.short

Line 189-191: I repeat a comment from the previous version exactly: why would a speciation event change the evolution of a paralog? A gene copy is in given individuals within a given population, and is not affected by the fact that another population is or is not becoming a new species.

There is no evidence for adaptation in the ms, remove the term from the abstract.

Neofunctionalisation is not presented in Ohno 1970, it was introduced by Force et al 1999 (Genetics April 1, 1999 vol. 151 no. 4 1531-1545).

Experimental design

The authors continue to use a simple in-house algorithm to label tree nodes as duplication or speciation, whereas this is an active domain of research with many elaborate methods and known issues. I repeat my suggestion of using labels from Ensembl Compara. Other methods can provide such labels, see https://questfororthologs.org/orthology_databases

The Figure S2 is extremely important. dS is clearly saturating, and older dN/dS values are thus unreliable. This must be addressed at the very begining of the discussion, and all results must be interpreted in light of this observation.
Notably:
the difference between within-species (on average younger) and between-species (on average older, thus dS saturated) paralogs

Validity of the findings

The authors continue to write that asymmetry implies neo-functionalisation, but this is not true. There is nothing in sub-functionalisation which implies symmetry.

Additional comments

This manuscript by David et al presents an interesting test of the ortholog conjecture using dN/dS (ω) value changes in gene trees. I have previously reviewed this manuscript for another journal, and it has been improved in several ways. Yet I still have some comments.

Improvements include taking into account the bias in which gene trees have more duplications, by removing speciation-only tree and by randomising inside trees, and clarifying terminology in several places.

---

## Round 0.2 · Minor Revisions

Thank you for the efforts made in the revisions to your manuscript. The reviewers have all seen the revised manuscript again and are in large-part content with the revisions. There are just a couple of places where one reviewer requests some minor revision to the text to try to accommodate a more nuanced viewpoint or interpretation of the results. If these minor revisions to the text could be accommodated then there will be no obstacles to accepting this manuscript for publication

Reviewer 1 ·

Basic reporting

No comment

Experimental design

No comment

Validity of the findings

Review of David, Oaks, and Halanych: “Patterns of gene evolution following duplications and speciations in vertebrates.”

Overview:
In this manuscript, the authors describe having inferred a large set of gene trees that include branchings due both to speciation and duplication events. They divide the branches in this fashion and estimate branch-specific values of the selective constraint (Ka/Ks) for all branches. They then explore the pattern of shifts in Ka/Ks after speciation and duplication events. They argue that the pattern of increases in Ka/Ks after duplication events is consistent with the orthology conjecture: that orthologs maintain more similar functions over time than do gene duplicates.

Major comments:
The authors have generally responded to my prior concerns, but I still think the manuscript overstates its case in a couple of places:

Lines 45&46: I do not agree that selection for higher dosage, escape from adaptive conflict and selection to maintain dosage balance can or should be squeezed under a neofunctionalization/subfunctionalization dichotomy (1).

Lines 171 and following: As another reviewer also pointed out, an asymmetry in selective constraint between duplicates is not solely explicable through neofunctionalization but is also compatible with, at a minimum, subfunctionalization. The authors response here is not compelling: A duplication of a broadly expressed gene where one copy retains expression in most tissues and the other is expressed only in a few or at lower levels could quite easily generate the pattern the authors describe in the response. The reason is that in many cases the dominate force in shaping omega is not protein structure per se but expression level: see(2) This section needs to be revised.



References:
1. Hahn MW (2009) Distinguishing among evolutionary models for the maintenance of gene duplicates. Journal of Heredity 100(5):605-617.
2. Drummond DA, Raval A, & Wilke CO (2006) A single determinant dominates the rate of yeast protein evolution. Molecular Biology and Evolution 23(2):327-337.

Additional comments

No comments.

Reviewer 2 ·

Basic reporting

The authors have properly addressed our points.

Experimental design

Na

Validity of the findings

Na

Additional comments

Na

·

Basic reporting

No specific comments

Experimental design

No specific comments

Validity of the findings

No specific comments

Additional comments

The authors have answered all reviewer comments to my satisfaction. This is a nice addition to the literature on the ortholog conjecture and on patterns of protein selection.

---

## Round 0.3 · accepted · Accept

Thank you for the care with which you have addressed all of the reviewers' comments and your level of engagement with this process.